

# Photocatalytic degradation of Dyes in Water by Analytical Reagent Grade Photocatalysts – A comparative study

Dnyaneshwar R. Shinde, Popat S. Tambade*, Manohar G. Chaskar, Kisan M. Gadave

Prof. Ramkrishna More Arts, Commerce and Science College, Akurdi, Pune-44,

Affiliated Savitribai Phule Pune University (India)

*Correspondence to* : Popat S. Tambade (pstam3@rediffmail.com)

**Abstract:** In a search of low cost photocatalyst for dye degradation we have evaluated photocatalytic activity of the Analytical
Reagent (AR) grade ZnO, $TiO_2$ and $SnO_2$. The photocatalytic activity was evaluated towards the decolourization of structurally
diverse dyes such as crystal violet, basic blue and methyl red under solar irradiation and compared with benchmark Degussa
P-25 ($TiO_2$) photocatalyst. The received metal oxides were characterized by the different physicochemical methods of analysis.
Powder XRD analysis showed that these metal oxides are polycrystalline in nature and crystallized in different crystalline
phases. The crystalline phases of these oxides were found to be hexagonal for ZnO, tetragonal for $TiO_2$ and rutile for $SnO_2$.
Particle morphology was analysed through SEM imaging and it showed that these oxides consists of different particle
morphologies and have different particle sizes. Band gap was evaluated from diffuse reflectance spectra and it was found to
be 3.24, 3.20 and 3.66 eV respectively for ZnO, $TiO_2$ and $SnO_2$. Among the three AR grade oxides, ZnO exhibited highest
photocatalytic activity which is even higher than Degussa P-25 ($TiO_2$) photocatalyst. About 20% enhancement in the
photocatalytic activity of AR grade ZnO was observed when silver metal loaded of on ZnO surface.

**Keywords:** photocatalysis, dyes, decolourization, solar irradiation, metal oxide

## 1 Introduction

Textiles, paper, plastics, leather, food and cosmetic industries widely use dyes and pigments to color products.
Extensive disposal of dyes from these industries to water bodies poses great threat to aquatic life as well as environment.
Numerous studies have been undertaken to investigate the efficient removal of hazardous organic molecules from the
wastewater during the past several decades. These studies have demonstrated that the hazardous organic molecules in
wastewater can be decomposed by photoexcited charge carriers in an excited semiconductor (Yang et. al., 2001; Qi et. al.,
2014; Singh et. al., 2015; Wang et. al., 2016; Keane et. al., 2014; Shinde et. al., 2015). This process is termed as photocatalysis.
The most promising natural energy source for photocatalysis based on semiconductors is the sunlight (Nguyen et. al. 2015).
However, an optimal use of the sunlight lies in designing an efficient sunlight-driven photocatalyst system. Nowadays, many
semiconductors such as tin oxide ($SnO_2$), tungsten oxide ($WO_3$), titanium oxide ($TiO_2$), cerium oxide ($CeO_2$), and zinc oxide



(ZnO) have been applied in heterogeneous photocatalysis (Xu et.al., 2013; Herrmann, 1999). The most of these semiconductors are wide band gap semiconductors, thus promote photocatalysis by UV radiation of the electromagnetic spectrum (Lin and Lin, 2007; Abo et. al., 2016). When such a photocatalyst is irradiated with light of energy equal to or higher than its band gap energy, electron-hole pairs are created. In an aqueous medium, reactants may be adsorbed on the surface of photocatalyst and

react directly or indirectly with the photogenerated electrons and holes (Girish Kumar and Koteswara Rao, 2015; Nagaraja et. al., 2012). This phtotocatalysis process has demonstrated its efficiency in degrading a wide range of organic pollutants, hazardous inorganic materials, and microbial agents into readily biodegradable compounds, and eventually mineralized them to harmless carbon dioxide and water (Chong et. al., 2010). In the category of semiconductor photocatalysts $TiO_2$ and ZnO have been widely investigated (Pardeshi and Patil, 2008; Parida and Parija, 2006). $TiO_2$ is generally considered to be a non-

toxic, chemically inert and photostable catalyst and has the ability to degrade dyes as well as a number of organic pollutants (Hu et. al., 2013; Zhou et. al., 2011; Akpan and Hameed, 2009). Degussa P-25, a blend with both anatase and rutile $TiO_2$ is widely used as commercialized photocatalyst (Hou et. al., 2015). Degussa P-25 shows significant photocatalytic activity towards organic dye degradation as compared to other commercial forms of $TiO_2$ (Zhou et. al., 2012; Lydakis-Simantiris et. al., 2010). However extensive use of $TiO_2$ for large scale water treatment is uneconomical, thereby suitable alternatives to

$TiO_2$ have been searched worldwide.

ZnO has been broadly studied as a photocatalyst in water treatment due to its low cost and its good optoelectronic and catalytic properties (Nagaraja et. al., 2012). Many attempts have been made to study photocatalytic activity of different semiconductors such as SnO2, ZrO2, CdS and ZnO (Sakthivel et. al., 2003; Pawar et. al., 2016; Cheng et. al., 2011). ZnO is an n-type semiconductor with many attractive features. Zinc oxide has a wide band gap of 3.17 eV with a large number of

active sites and exhibits very high surface reactivity as compared to $TiO_2$ (with anatase band gap 3.2 eV). ZnO is an efficient visible light photocatalyst and capable to generate hydroxyl radicals in sufficient quantity. Surface and core defects like oxygen vacancies, zinc interstitials, oxygen interstitials in ZnO play a vital role in photocatalysis reactions by providing active sites to prevent electron and holes recombination (Hans et. al., 2012, Janotti and Van de Walle, 2009). This in turn enhances the generation of hydrogen peroxide ($H_2O_2$), superoxide ($O_2^{-*}$) and hydroxyl ($OH^*$) radicals on the ZnO surface and are reported

to be responsible for the photocatalytic activity (Girish Kumar and Koteswara Rao, 2015). There are reports that doping of ZnO nanoparticles with metal and transition metals like Ag, Pb, Mn, and Co can increase the photocatalytic activity as doping increases the surface defects (Patil et. al., 2010; Sood et. al., 2014, Zhi-gang et. al., 2012).

We have systematically studied the photocatalytic activity of commercial AR grade ZnO, $TiO_2$, and $SnO_2$ powders using controlled degradation towards the degradation of industrial dyes crystal violet (CV), basic blue (BB) and methyl red

(MR) in aqueous solution under solar irradiation. Activity of these photocatalysts were compared with Degussa P-25 photocatalyst. Flat slurry reactor was used for the study of photocatalytic activity. Then ZnO was loaded with Ag and effect on photocatalytic activity was studied. We hereby propose that this work will be significant for a scientific as well as for a practical application in the treatment of wastewater caused by the widespread disposal of organic pollutants.



## 2 Experimental methods

### 2.1 Materials

Degussa P-25 photocatalyst was obtained from Nanoshel (US) (Characteristic information: Support file Table-1) and used as benchmark photocatalyst. Commercial $TiO_2$, $ZnO_2$ and $SnO_2$ were obtained from Loba Chemie Ltd. (India). Three structurally diverse dyes viz., crystal violet (CV), methyl red (MR) and basic blue (BB) used in this investigation were purchased from Sigma-Aldrich (India). All the materials were used without further purification. Tap water was used for preparation of dye solutions. pH of the solutions was adjusted using 1 M NaOH or 1 M HCl.

### 2.2 Characterization of Photocatalysts

The powder X-ray diffractogram (XRD) of as received ZnO, $TiO_2$ and $SnO_2$ were recorded and analysed for their crystal structure and lattice parameters. The average particle size of each oxide was estimated by the Debye-Scherrer formula. The band gap was evaluated from diffuse reflectance spectra (DRS) of metal oxides in absorbance mode by prescribed method. Particle morphology was analysed by the scanning electron microscopic (SEM) imaging. Specific surface area was obtained by the prescribed method (Jiji et. al., 2006; Jo-Yong et. al., 2006).

### 2.3 Instruments

Photocatalytic activity was carried out in specially designed flat slurry reactor (FSR). The reactor consists of rectangular shaped reaction vessel of size $20 \times 30 \times 4$ cm$^3$. The depth of solution maintained in the vessel is ~1.5 cm. The vessel is attached to stirring tank (volume 300 ml) through collection and pumping line (Fig.1). Constant stirring of solution was insured using stirrer to keep the mixture in suspension. The height of dye solution in reaction vessel was maintained at 1 cm and the dye solution was re-circulated through the stirring tank. The experiments were performed in sunlight between 10 am to 3 pm. During degradation reaction the absorbance of the dye solution was recorded at constant interval of time.

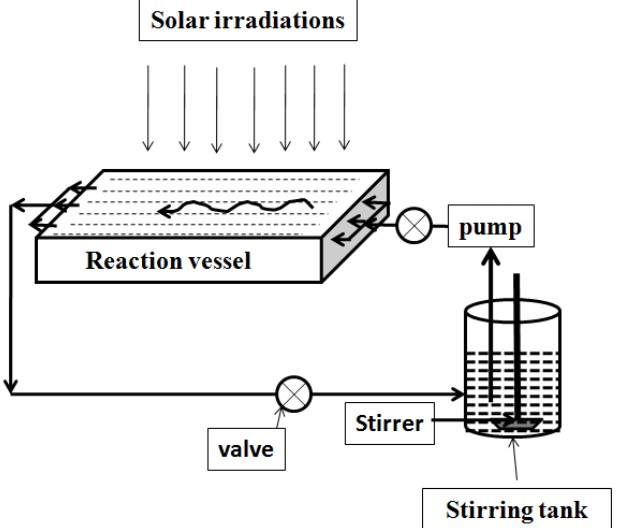

**Fig.1** The reactor set up for photodegradation of dye under sunlight

### 2.4 Photocatalytic experiments



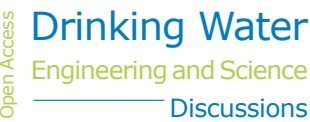

Separate dye solutions of CV, MR and BB were prepared in a tap water with a concentration of 10 ppm. To each 2000 ml dye solution, photocatalyst (400 mg L$^{-1}$) was added and suspension was developed. The suspension were stirred in the dark for 1 h to reach the adsorption-desorption equilibrium before exposure to the sunlight. Then the suspension was transferred to the reactor for photocatalytic activity and suspension was subjected to solar irradiation. The same procedure was used for all the catalysts used in the study. The photocatalytic activity was determined under optimized condition of pH of the dye solution (pH=9). At certain time intervals a small quantity of solution was taken out from the reactor with the help of a syringe. Then absorbance were recorded at $\lambda_{max}$ of the dyes and rate of decolorization was observed in terms of rate constant. The rate constant was determined using Langmuir-Hinshelwood equation (Chong et. al., 2009):

$$k = \frac{2.303}{t} log_{10}\left(\frac{A_0}{A}\right) \qquad\qquad ...(1)$$

where k is the apparent reaction rate constant, $A_0$ is the initial absorbance of dye solution, and A is the absorbance of dye solution after irradiation at time t.

**2.5 Loading Ag metal on AR grade ZnO**

5 g AR grade ZnO was sonicated for five minutes in 100 mL 0.01M AgNO$_3$ solution, stirred for 30 minutes and centrifuged to recover ZnO. The recovered ZnO was suspended in alkaline solution (with ammonia) of glucose (0.1M, 100 ml, pH=9) and boiled for 5 min. and ZnO was recovered by centrifugation; it was washed with distilled water till free from alkali. Finally, it was annealed for 30 min. in the furnace at 450 °C.

**3 Results and discussion**

The photocatalytic experiments were carried out under solar light. Different catalysts viz., ZnO, TiO$_2$, SnO$_2$ and P-25 were investigated for their decolorization rate constant. The rate constant of decolorization was recorded in terms of change in absorbance of characteristic peaks of the dyes.

**3.1 Characterisation of Catalysts**

The powder XRD analyses were conducted on the as received AR grade ZnO, TiO$_2$ and SnO$_2$ to investigate their crystalline phases and average crystallite sizes. Fig. 2 represents XRD peaks of the as received three semiconductor samples.

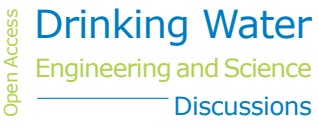

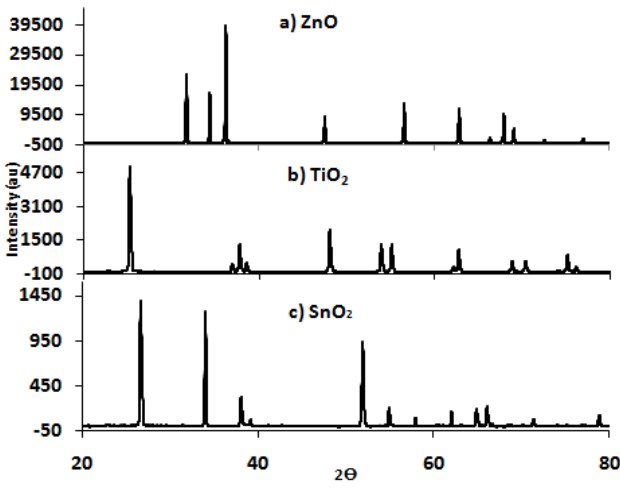

**Fig. 2** XRD of a) ZnO, b) $TiO_2$ and c) $SnO_2$

Fig. 2(a) presents the XRD patterns of ZnO. The major diffraction peaks are observed for ZnO at 2Θ angles 31.78°,
34.44°, 36.27°, 47.55°, 56.61°, 62.87°, 66.39°, 67.96°, and 69.10° which are indexed to the planes [100], [002], [101], [102],
[110], [103], [200], [112] and [201] respectively. These peaks are matching with hexagonal wurtzite structure of ZnO (JCPDS
36-1451) (Pawar et. al., 2016). The separate and sharp diffraction peaks in Fig. 2(b) are found at 25.3°, 37.79°, 48.04°, 53.9°,
55.07°, 62.7°, 68.78°, 70.30°, and 75.08°, corresponding to [101], [004], [200], [105], [211], [213], [116], [220] and [215]
planes of pure anatase phase of $TiO_2$ (JCPDS 21-1272) respectively. Fig. 2(c) presents diffraction peaks at 26.66°, 34.03°,
38.07°, 39.15°, 42.67°, 51.93°, 54.94°, 57.96°, 62.04°, 64.88°, 66.07°, 71.34°, and 78.81° which can be indexed to [100],
[101], [200], [111], [220], [211], [220], [002], [310], [112], [301], [202], and [321] planes of rutile (cassiterite) crystalline
phase of $SnO_2$ (JCPDS 77-0451) (Zhang et. al., 2009, Elsayed et. al., 2014).

Lattice parameters were calculated for all three metal oxides from XRD data and are listed in Table 1. These were
found in good agreement with reported values (Pawar et. al., 2016; Cheng et. al., 2011; Li et. al., 2014; Li et. al., 2013).
Crystallite size was calculated from the intense peaks in XRD by Debay-Sherrer formula and are found to be 103 nm, 36 nm
and 66 nm for ZnO, $TiO_2$ and $SnO_2$ respectively.

**Table 1.** Unit cell parameters calculated from XRD data

| Parameters | ZnO | $TiO_2$ | $SnO_2$ |
|---|---|---|---|
| a (Å) | 3.248 | 3.784 | 4.734 |
| c (Å) | 5.203 | 9.514 | 3.179 |
| c/a ratio | 1.602 | 2.514 | 0.672 |
| Unit cell volume ($\times 10^{-23}$ cm$^3$) | 4.756 | 13.62 | 7.125 |
| X-ray Density (g cm$^{-3}$) | 5.681 | 3.894 | 7.023 |

## 3.2 SEM micrographs of catalysts

Fig. 3 shows typical SEM micrographs of as received catalysts.




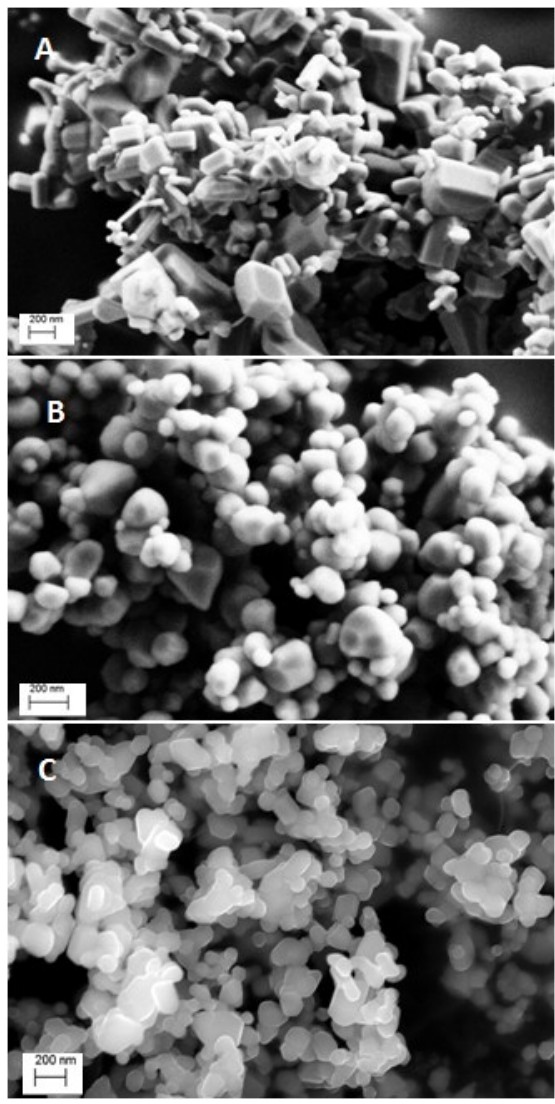

**Fig. 3** SEM micrographs of (A) ZnO, (B) TiO$_2$ and (C) SnO$_2$

  The SEM micrographs of metal oxides show the agglomeration of primary particles (crystallites) to form grains of larger size. ZnO crystallites were agglomerated to less extent into cubic shaped grains, while to large extent in elongated hexagonal rods of variable diameter. TiO$_2$ grains are roughly spherical in shape of variable diameter from 55 to 170 nm. SnO$_2$ shows roughly spherical or irregular shape particles of size ranges from over 86 to 230 nm. The particles of all three oxides

10  are tiny and have smooth surfaces.



### 3.3 UV–vis absorption spectra of catalysts

The corresponding UV-vis absorbance spectra of as received catalysts viz., ZnO, $TiO_2$ and $SnO_2$ are provided in Fig. 4. The UV-vis spectra of these metal oxides show strong absorbance in UV light region and an absorption edge between 300 to 400 nm owing to the relatively large excitation binding energy.

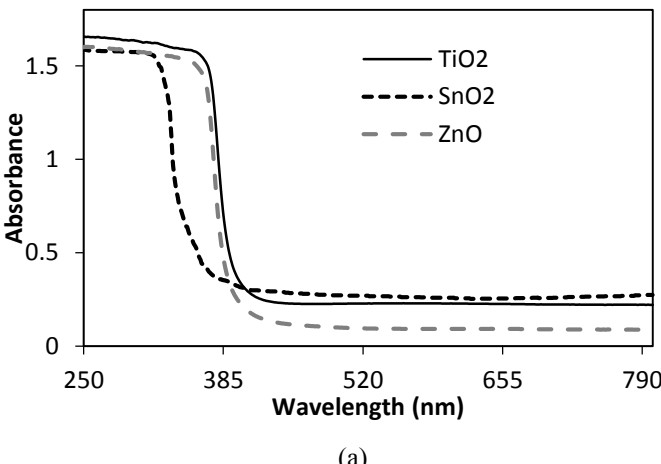

(a)

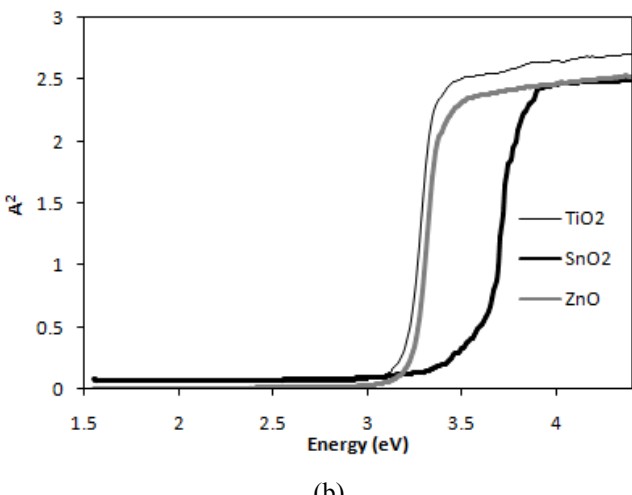

(b)

**Fig. 4**. (a) Diffuse absorbance spectra of ZnO, $TiO_2$ and $SnO_2$, (b) Graph $A^2$ against energy constructed from UV-DRS in absorbance mode to evaluate band gap

The absorbance onsets of wurtize ZnO, anatase $TiO_2$ and rutile $SnO_2$ are 384 nm, 387nm and 341 nm respectively. The experimentally evaluated band gaps for ZnO, $TiO_2$ and $SnO_2$ are found to be 3.23, 3.20 and 3.64 eV respectively (Support file, Fig.1). The observed band gaps are in good correlation with reported values (Cheng et. al., 2013; Anand et. al., 2008; Gupta and Tripathi, 2011). The band gap order of these three oxides is $TiO_2 < ZnO < SnO_2$.



### 3.4 Specific Surface area of catalysts

Specific surface areas of AR grade ZnO, TiO$_2$ and SnO$_2$ powders were determined by prescribed method. Among these three oxides TiO$_2$ exhibited highest specific surface area (42.54 m$^2$g$^{-1}$) owing to its smallest particle size and low crystal density. ZnO and SnO$_2$ have comparable specific surface areas 10.183 and 12.91 m$^2$g$^{-1}$ respectively, which are approximately four times less than TiO$_2$. Such difference can be attributed to large crystallite size and higher crystal densities of ZnO and SnO$_2$ powders. The specific surface area of Degussa P-25 catalyst is ~50 m$^2$g$^{-1}$.

### 3.5 Photocatalytic Activity

In order to compare the photocatalytic activity of commercially available photocatalysts viz. ZnO, TiO$_2$, SnO$_2$ and P-25, a set of photocatalytic experiments were performed under similar conditions using CV, BB and MR dyes. Initially blank experiments were performed under solar irradiation without addition of any catalyst to the dye solutions and negligible decolourization was observed. The photocatalytic activities are compared in terms of rate constant of decolourization of dyes which were obtained using the graphs of $\log_{10}(A_0/A_t)$ against time (Support File Fig. 1) on photocatalysts and are listed in Table 2.

**Table 2.** Comparison of photocatalytic activities in terms of reaction rate constants for three different dyes

| Catalyst | CV | | BB | | MR | |
|---|---|---|---|---|---|---|
| | k ( min$^{-1}$) | t$_{1/2}$ (min) | k ( min$^{-1}$) | t$_{1/2}$ (min) | k ( min$^{-1}$) | t$_{1/2}$ (min) |
| ZnO | 0.079 | 8.8 | 0.1 | 6.9 | 0.014 | 49.6 |
| TiO$_2$ | 0.026 | 26.9 | 0.045 | 15.5 | 0.008 | 84.1 |
| SnO$_2$ | 0.010 | 71.0 | 0.017 | 40.7 | 0.004 | 196.7 |
| Degussa P-25 | 0.060 | 11.6 | 0.076 | 9.1 | 0.012 | 56.4 |

The results indicated that for all three dyes ZnO exhibits higher photocatalytic activity than TiO$_2$, SnO$_2$ and P-25. Comparative studies on ZnO and TiO$_2$ (Degussa P-25 TiO$_2$) photocatalysts was performed by many other researchers where ZnO showed larger performance (Hussein and Abass, 2010; Sakthivel et. al., 2003; Qamar and Muneer, 2009). Han et al. (2012) accounted enhanced photocatalytic activity of ZnO in terms of higher absorbance of radiations by ZnO over TiO$_2$ in ultraviolet region. Surface area of the photocatalyst is one of the key parameters affecting photocatalytic activity. Among the four catalysts used in the study, ZnO displayed the lowest specific surface area while Degussa P-25 has the highest specific surface area. The results showed that despite of having low specific surface area ZnO exhibits higher photocatalytic activity. This might be due to different intrinsic characteristics of ZnO than TiO$_2$ and SnO$_2$. *This may be explained on the basis of quantum efficiency as ZnO have greater quantum efficiency than TiO$_2$ and SnO$_2$* (Kansal et. al., 2007). Another important factor affecting the photocatalytic activity is the band gap. SnO$_2$ exhibits less activity because of their wide band gap (3.64 eV) and need a large amount of UV radiation to excite electron–hole pairs in this catalyst (Sakthivel et. al., 2003; Abo et. al., 2016). ZnO and TiO$_2$ have comparable band gaps (3.24 and 3.20 eV respectively). Degussa P-25 have the lowest band gap (3.00 eV) and expected to show higher photocatalytic activity than ZnO. However, experimental results clearly indicated that



ZnO has higher photocatalytic activity than Degussa P-25 and TiO$_2$. Sunlight consists of 5-7% radiations in the ultraviolet region. The results of photocatalytic activity provides a indirect evidence that ZnO may be able to absorb larger fractions of photon energy more efficiently than P-25 and TiO$_2$ within the UV region (Pardeshi and Patil, 2008). This may be attributed to intrinsic defects in ZnO crystals. The predominant defects in ZnO are positively charged Zn interstitials and oxygen vacancies. When electron-hole pairs are formed by photo-excitation, it is suspected that these Zn interstitials and oxygen vacancies facilitate redox reactions by trapping photo-generated electrons. This reduces the recombination of electrons and holes and they are available for dye degradation (Hans et. al., 2012). Degussa P-25 catalyst is a blend of rutile (30%) and anatase (70%) TiO$_2$ which prevents the recombination of electrons and holes and due to this it shows more photocatalytic activity than anatase TiO$_2$. Since the rutile phase is reportedly less effective than anatase in photocatalysis , P-25 shows less activity than ZnO.

Dye degradation by the photocatalyst under sunlight takes place by two different ways. In the first mechanism, when the photocatalyst is illuminated, a photon of energy higher or equal to the band gap, causes excitation of electrons into the conduction band (CB) of photocatalyst. Simultaneously, an equal number of holes are generated in the valence band. The high oxidative potential of the hole in the valance band of catalyst permits the direct oxidation of the dye to reactive intermediates followed by degradation. The process is expressed as (Konstantinou, and Albanis, 2004):

$$MO/MO_2 + h\upsilon \rightarrow MO/MO_2(e_{CB}^- + h_{VB}^+) \qquad \text{…(2)}$$

$$h_{VB}^+ + dye \rightarrow dye^* \rightarrow dye\ degradation \qquad \text{….(3)}$$

The degeneration of dye also occurs due to the reaction with hydroxyl ($OH^\bullet$) and superoxide ($O_2^{-*}$) free radicals. OH$^*$ is a highly reactive strong oxidizing chemical species is either formed by decomposition of water or reaction of hole with surface-bound hydroxyl groups ($OH^-$):

$$h_{VB}^+ + H_2O \rightarrow H^+ + OH^* \qquad \text{…(4)}$$

$$h_{VB}^+ + OH^- \rightarrow OH^* \qquad \text{…(5)}$$

The conduction band ($e_{CB}^-$) potential was negative enough to reduce O$_2$ which is adsorbed on the surface of catalyst to generate $O_2^{-*}$.

$$e_{CB}^- + O_2 \rightarrow O_2^{-*} \qquad \text{…(6)}$$

The subsequent reactions of $OH^-$ and $O_2^{-*}$ can also generate $OH^\bullet$. The reactions of $OH^\bullet$ and $O_2^{-*}$ with dye molecules degrade their moiety completely into simple molecules (Chen and Ray, 2001).

Second mechanism is a dye sensitization mechanism. In this mechanism, the dye molecules adsorbed on the surface of catalyst absorb visible radiations and undergoes electronic excitation from highest occupied molecular orbitals (HOMO) to lowest unoccupied molecular orbitals (LUMO). Then excited electron from LUMO of the dye molecule is injected to conduction band (CB) of the photocatalyst and the dye is converted into cationic dye radical (Lakshmi Prasanna and Rajagopalan, 2016). This dye radical undergoes degradation to produce mineralized products (Li et. al., 2002; Muthirulan et.



al., 2013). The electron in CB of metal oxide further scavenged by oxygen to produce $O_2^{-*}$ and resulted into decolourisation of dyes. This mechanism is prominent when dye molecules are in adsorbed state on the catalyst surface. These two mechanisms are depicted in Fig. 5.

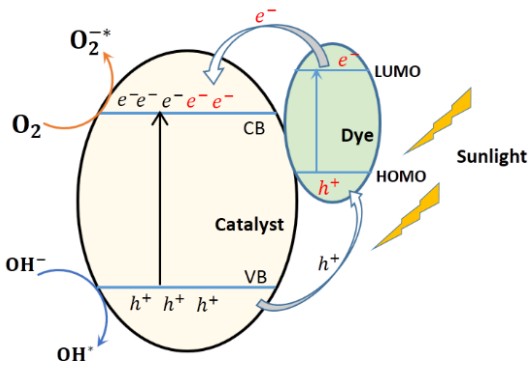

$$OH^*/O_2^{-*} + Dye \rightarrow Dye\ degradation$$

**Fig. 5. Schematics of photodegradation of dye in sunlight**

The adsorption study of dyes in solution state by photocatalyst showed that CV and BB were adsorbed to certain extent on all four photocatalyst while MR was adsorbed to negligible extent (Table 3).

**Table 3.** Percent adsorption of three dyes on photocatalysts

| Catalyst | Percent adsorption | | |
|---|---|---|---|
| | **CV** | **BB** | **MR** |
| ZnO | 9.8±1.2 | 16.3±0.8 | negligible |
| TiO$_2$ | 5.3±0.7 | 8.4±1.1 | negligible |
| SnO$_2$ | 4.6±0.8 | 7.4±0.4 | negligible |
| Degussa | 6.9±0.6 | 7.4±0.6 | negligible |

The quantity of adsorption clearly reflected in terms of the decolourization rate constants of these dyes. When rate constants represented in Table 1 are compared with percentage adsorption (Table 3), it is observed that MR has the least rate constant because of negligible adsorption. Furthermore, BB is adsorbed to greater extent than CV thereby it showed greater decolourization rates with all four catalysts. This supports the dye sensitized mechanism. The results of this study are in agreement with the earlier findings (Han et. al., 2012; Kansal et. al., 2007), that ZnO can harvest maximum solar energy by utilizing visible light for decoularization of dye in water.

**3.6 Enhancement of photocatalytic activity of AR grade ZnO**

To broaden the photoresponse of ZnO catalyst for solar spectrum, various material engineering approaches have been devised. These approaches include doping (with metal, non-metal or metalloid), composite with carbon nanotubes, formation



of hetero-structures with noble metals or other semiconductors, dye sensitizers (Zhang and Zeng, 2010) . Herein, an attempt had been made to enhance photocatalytic activity of commercial ZnO by loading noble silver metal on its surface. The suspended ZnO in AgNO₃ adsorb Ag⁺ ions on its surface (adsorbed quantity 1.9 mg per gram ZnO). When ZnO consisting of adsorbed Ag⁺ was treated with alkaline glucose solution then Ag⁺ get converted to silver metal by chemical reaction (Eq. 7).

$$Ag^+ + C_6H_{12}O_6 \xrightarrow{NH_4OH} Ag_{metal} + C_6H_{12}O_7 \quad \text{.... (7)}$$

The formation of Ag metal on surface is clearly indicated by faint grey colour of surface sensitized ZnO. The Ag metal sensitized ZnO was characterized by powder XRD which showed extra peaks at 2Θ=38.1°, 44.3°, 64.5° and 77.9° in addition to hexagonal wurtzite peaks of ZnO (support file, Fig-3). These are the characteristics peaks of FCC crystallized Ag metal (JCDS data file no. 04-0783) (Ma et. al., 2014). Smaller peak intensities corresponding to Ag metal indicate its smaller crystal

10   size on the surface of ZnO. It is also confirmed by the EDS analysis and DRS in absorbance mode (support file, Fig-3 and 4).

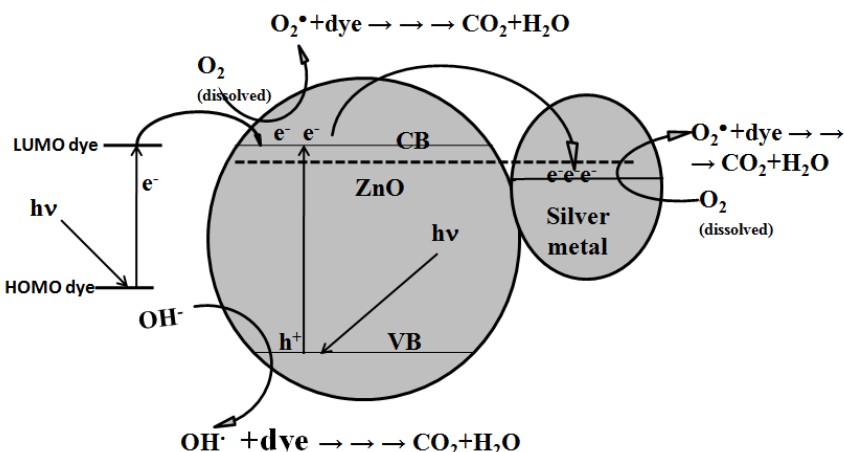

**Fig. 6** Schematic diagram representing the band structure silver metal loaded ZnO, electronic excitation process, dye sensitized mechanism and electron delocalization in silver metal loaded ZnO

The process of phtotocatalysis by Ag/ZnO is depicted in Fig. 6. The coupling of Ag metal with ZnO alters the band structure of the photocatalyst. The CB energy of Ag metal is lower than CB of ZnO. Molecules of the dyes which are adhered to the ZnO particle surface are excited by the incoming light release electrons from LUMO to the CB of ZnO. The ZnO is also excited by UV component of sunlight, forming electron-hole pairs and electrons are available in the CB of ZnO. Eventually,

20   these electrons are transported to the CB of the Ag metal. Such a type of delocalization of the electrons is thermodynamically favourable and help to reduce rate of recombination of electron-hole pair produced on photo-excitation of ZnO. In turn, more electrons and holes are made available for chemical reaction which facilitates to enhance photocatalytic activity of Ag metal loaded ZnO (Kuriakose et.al., 2014; Jing et. al., 2006). These electrons and holes are responsible for the generation of $O_2^{-*}$ and

$OH^{\bullet}$ free radicals respectively, which act as extremely strong oxidants for the decomposition of dyes.

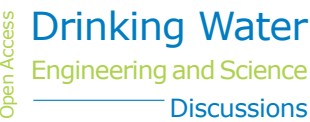

The calculated $k$ for the Ag loaded ZnO nanostructure is 0.017 min$^{-1}$ for MR dye which is 1.22 times higher than pure ZnO. For CV and BB the calculate k is 1.28 and 1.33 times higher than pure ZnO respectively.  These results indicate that the loading of Ag on ZnO enhances the photocatalytic activity of ZnO compared to pure ZnO in sunlight. It has been reported that Ag deposits on ZnO surface acted as electron sinks and hindered the recombination of photoinduced electrons and holes. This had ensured better charge separation than ZnO. The charge-transfer process occurring at Ag-ZnO interface assists the transport of electrons to the surface, which is responsible to enhance the photocatalytic activity (Patil et. al., 2015, Zeng et. al., 2008). Dinesh et. al. (2014) showed that the hybrid ZnO@Ag core-shell nanorods exhibit good dye degradation property as compared to the ZnO nanorods towards the degradation of Rhodamine 6G, Congo red and Amido black 10B dyes].

### 3.7 Cost comparison

The suitability of employing of particular photocatalyst in photocatalytic degradation of organic pollutants in the water is governed by its activity and cost. The costs of ZnO, TiO$_2$ and SnO$_2$ per kg are Rs. 1100, Rs. 2800 and Rs, 8000 respectively, while the cost of Degusa P25 is Rs.1,20,000/- per kg. The comparison of costs reveals that AR grade ZnO is less expensive and is affordable than Degussa p-25, AR grade TiO$_2$ and SnO$_2$. Even though, accounting the cost of staring material and other chemicals, manpower, time, special requirement of equipments, and electricity consumed, any of the laboratories synthesized version of ZnO, TiO$_2$ and SnO$_2$ will be costly than the commercial ones. In brief, AR grade ZnO turns out to be a cost effective and efficient material for photocatalytic removal of organic dyes under solar irradiation from the aqueous medium.

### 3.8 Future plan

We are presently working on other issues of ZnO photocatalysis such as an enhancement of stability and photocatalytic activity, recovery and reutilization of ZnO photocatalyst, we are also working on the project of optimization of reactor design of FSR to pilot scale.

### 4. Conclusion

The comparison of photocatalytic activity of ZnO with the benchmark, Degussa p-25 (TiO$_2$) catalyst as well as with AR grade TiO$_2$ and SnO$_2$ showed that despite of low specific surface area and larger grain size, ZnO displayed higher photocatalytic activity towards the destruction of industrial dyes (CV, BB and MR) under solar irradiation. The enhancement of photocatalytic activity of AR grade ZnO is possible by coupling it with noble metal like silver.  AR grade ZnO has low cost than other photocatalyst used in the study which make it strong candidate as a photocatalyst for dye degradation from aqueous medium under solar irradiation.

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
