# Peer review of "Photocatalytic degradation of Dyes in Water by Analytical Reagent Grade Photocatalysts – A comparative study"

_Drinking Water Engineering and Science, 2017_

## Referee Comment (RC1) · Anonymous Referee #1 · 15 Jul 2017

Manuscript dwes-2017-20 Title: Photocatalytic degradation of Dyes in Water by Analytical Reagent Grade Photocatalysts – A comparative study Authors: Dnyaneshwar R. Shinde, Popat S. Tambade*, Manohar G. Chaskar, Kisan M. Gadave In order to find efficient and low cost photocatalyst for dye degradation in aqueous solution, this paper evaluates photocatalytic activity of the Analytical Reagent (AR) grade ZnO, TiO2, SnO2, and Degussa P-25. The Photocatalytic experiments were carried out in a specially designed flat slurry reactor (FSR) and determined under optimized condition of pH of the dye solution (pH=9), different kinds of photocatalysts are used to degradate industrial dyes crystal violet (CV), basic blue (BB) and methyl red (MR) in aqueous solution under solar irradiation. In addition, the structures and characters of all the catalysts were evaluated by XRD, SEM, UV-vis spectrometer, and so on. The topic of the manuscript is interesting. However, the manuscript is not well written and organized due to poor English. Several illogical express and many editorial mistakes can be found in the paper. Therefore, the manuscript has to be refined due to concerns cited below.

Major comments: Technical comments 1)Page1.The abstract does not cover the main points of the manuscript. In the article, the photocatalytic activity experiments and results are foremost. I strongly suggest the author to rewrite it with logical organization. 2)Page 10ïijŐYou mentioned that "When rate constants represented in Table 1 are compared with percentage adsorption (Table 3), it is ...", Table 1 is XRD data, so the comparing is meaningless. Please revise it. 3)The photolytic degradations of dyes as control are not provided, which is very important for the mechanism analysis. For example, photosensitization is related with the light intake and photolytic degradation of dyes. 4)The pH conditions of the experiments are not clear. If the pH means a initial pH, the change of pH during the experiments should be provided. Otherwise, the buffer solution used should be clarified. Editorial comments Some, not all, of the editorial mistakes are shown below: 5)Page 1. 18th line .Should use "ZnO surface is loaded with silver metal" instead of "silver metal loaded of on ZnO surface". 6)Page 2. 24th line. Should use "is reported to be responsible for the photocatalystic activity " instead of "are reported to be responsible for the photocatalysitc activity". 7)Page 8. 26th line. Should use"and needs a large amount of UV radiation to excite electron–hole pairs in this catalyst " instead of"and need a large amount of UV radiation to excite electron–hole pairs in this catalyst ". 8)Page 8. 27th line. Should use"Degussa P-25 has the lowest band gap"instead of "Degussa P-25 have the lowest band gap". 9)Page 9. 8th line, "and due to this it shows more photocatalytic activity than anatase",sentence structure should be modificated. 10)page 11. 21th line, shoud use"and helps to reduce rate of .."instead of "and help to reduce rate of..".

---

## Referee Comment (RC2) · Anonymous Referee #2 · 20 Aug 2017

This study comparatively investigated the photocatalytic degradation of dyes by ZnO, TiO2 and SnO2 photocatalysts. The characteristics of such catalysts were also examined by XRD, SEM, etc. Overall, the study presents some new and interesting results, however, its significance and contribution in this research topic appear to be limited in terms of imcomplete experimental design and poor English. With these reasons, the publication of this work at the present form is not recommended in Drinking Water Engineering and Science (DWES). Significant revision and additional experimental works are necessary for further consideration of this work. 1. General: Please have a professional technical English editorial office to proof read the manuscript and unify units (e.g. ppm and mg/L). 2. The solar intensity variation with time and solar wavelength

spectra should be given. 3. Why the authors chose flat slurry reactor (FSR) instead of closed container, which was used in most of literatures? Was the water temperature in FSR maintained? 4. Page 4, Lines 5∼6: Was the solution pH maintained at 9? If not, the variation of pH value should be monitored. 5. Page 8, Line 15: The conclusion was hasty because the authors just investigated the photocatalytic activity at only one condition. Factors such as initial dye concentration, catalyst loading, irradiation time, pH and intensity of light should be considered. 6. TOC analysis was suggested to help study the photodegradation performance. 7. Please explain the significant difference of rate constants for three different dyes in Table 2. 8. Page 10, Lines 13∼14: "Table 1" should be corrected to "Table 2".

---

## Author Comment (AC2) · 16 Sep 2017

Respected sir, We are thankful for reviewing the article and suggesting fruitful modifications. In following section we answered the queries of both referees and appropriate explanations were also included in revised manuscript.

Answers to Comments: Referee-1

Q.1: Page1.The abstract does not cover the main points of the manuscript. In the article, the photocatalytic activity experiments and results are foremost. I strongly suggest the author to rewrite it with logical organization. Ans: The abstract is modified in the

revised manuscript according to suggestion given by the Referee.

Q.2. Page 10. You mentioned that "When rate constants represented in Table 1 are-compared with percentage adsorption (Table 3), it is : : :", Table 1 is XRD data, so thecomparing is meaningless. Please revise it. Ans. Table numbers and captions are corrected and arranged properly in the revised manuscript.

Q.3. The photolytic degradations of dyesas control are not provided, which is very important for the mechanism analysis. Forexample, photosensitization is related with the light intake and photolytic degradationof dyes. Ans: Control experiments were carried out simultaneously. All three dyes selected in the study were found to be stable to light and pH in the control experiment. In revised manuscript it is mentioned clearly. Page -7: line -18 ff, page-9: line 1-2.

Q.4.The pH conditions of the experiments are not clear. If the pH means ainitial pH, the change of pH during the experiments should be provided. Otherwise,the buffer solution used should be clarified Ans: pH was adjusted initially. The pH was monitored during the experiments andfound to be decreased up to 8.6±0.1. Explanation of this part is provided in revised manuscript page-3: line -20.

Q.5.Page 10. 18th line .Should use "ZnO surface is loaded with silver metal" instead of "silver metal loaded of on ZnO surface". Ans: It is corrected in revised manuscript. The revised manuscript is language corrected form professional language editors (editingindia - Scholarly Editing and Translation Services Pvt Ltd –India). Page-10: line-17 Q.6 Page 2. 24th line. Should use "is reported to be responsible for the photocatalytic activity" instead of "are reported to be responsible for the photocatalytic activity". Ans: It is corrected in revised manuscript. Pae-2: line-20.

Q.7.Page 8. 26th line. Should use and "needs a large amount of UV radiation to excite electron–hole pairs in this catalyst " instead of "and need a large amount of UV radiation to excited electron–hole pairs in this catalyst ". Ans: Corrections are made according to suggestion. Page-8: line-24. Q.8.Page 8. 27th line. Should use"Degussa

P-25has the lowest band gap" instead of "Degussa P-25 have the lowest band gap".
Ans: Corrections are made according to suggestion. Page-9; line-2 Q.9.Page9. 8th
line, "and due to this it shows more photocatalytic activity than anatase", sentence
structure should be modificated. Ans: Corrections are made according to suggestion.
The line is modified in language editing. Page-9: Line-10-11 Q.10. Page 11. 21th line,
should use"and helps to reducerate of "instead of "and help to reduce rate of..". Ans:
Corrections are made according to suggestion. Page-11: Line- 8- 9.

Answers to Comments : Referee-2

Q.1. General: Please have a professional technical English editorial office to proof read
the manuscript and unify units (e.g. ppm and mg/L). Ans: The revised manuscript is
language corrected form professional language editors (editing India - Scholarly Editing
and Translation Services Pvt Ltd –India). Corrections in units are made.

Q. 2. The solar intensity variation with time and solar wavelength spectra should be
given. Ans: time of the experiments and intensity of sunlight is mentioned clearly.
Wavelength spectrum of solar radiations is not recorded but it is available elsewhere
easily. Page-3, line-15,

Q.3.Why the authors chose flat slurry reactor (FSR) instead of closed container, which
was used in most of literatures? Was the water temperature in FSR maintained? Ans:
In present study we have utilized FSR, since it can be irradiated with external source of
radiation such as sunlight. It provides large surface area and uniform depth of reaction
mixture throughout the reactor.It is open to air hence provide more dissolved $O_2$ for
chemical reaction. Annular slurry reactor or similar type of design may not be utilized
in sunlight since they are internally irradiated and not suitable for external irradiation
sources. Tubular reactor can be used in sunlight but it provides less surface area and
uneven depth of reaction mixture in the vessel. Closed container reactor can provide
less aeration hence less dissolved $O_2$ for chemical reaction. It is mentioned page
page-3: line-21, page-7: line-15-16 Q. 4. Page 4, Lines 5-6: Was the solution pH

maintained at 9? If not, the variation of pH value should be monitored. Ans: pH was adjusted initially. During the experiment it was decreased up to 8.6 ±0.1. Page-3: line 20-21 Explanation of this part is provided in revised manuscript. Page -7: Line -18.

Q.5. Page 8, Line 15: The conclusion was hasty because the authors just investigated the photocatalytic activity at only one condition. Factors such as initial dye concentration, catalyst loading, irradiation time, pH and intensity of light should be considered. Ans: The conclusion part is revised according to suggestions.

Q.6. TOC analysis was suggested to help study the photodegradation performance. Ans: Our main aim of study is comparison of photocatalytic activity of metal oxides which is accounted in terms of the rate of decolourization. After decolourization sage COD was determined for the dye solutions treated with ZnO photocatalyst. For the dye solution treated with TiO2, Degussa P-25 and SnO2 COD analyses were not performed as these catalysts displayed low catalytic activity. Separate section is added in revised manuscript. Page-11: line-25 ff and Support file, Table-3.

Q. 7. Please explain the significant differenceof rate constants for three different dyes in Table 2. Ans: It is mentioned. Page-10: line-8. Q.8. Page 10, Lines 13_14: "Table1" should be corrected to "Table 2". Ans: Table numbers and captions are corrected and arranged properly in the revised manuscript.

Please also note the supplement to this comment:
https://www.drink-water-eng-sci-discuss.net/dwes-2017-20/dwes-2017-20-AC2-supplement.zip
* * *